# Scanning Electron Microscopy Reveals New Ultrastructural Features in Metacercariae of *Clinostomum cutaneum* (Digenea: Clinostomidae) Infecting *Oreochromis niloticus* (Actinopterygii: Cichlidae) in Kenya

**DOI:** 10.3390/pathogens14030249

**Published:** 2025-03-04

**Authors:** Miriam Isoyi Shigoley, Nikol Kmentová, Daniel Mungai Ndegwa, Martina Topić, Kelly J. M. Thys, Maarten P. M. Vanhove

**Affiliations:** 1Research Group Zoology: Biodiversity & Toxicology, Centre for Environmental Sciences, Hasselt University, Agoralaan Gebouw D, 3590 Diepenbeek, Belgium; nikol.kmentova@uhasselt.be (N.K.); martina.topic@uhasselt.be (M.T.); kelly.thys@uhasselt.be (K.J.M.T.); maarten.vanhove@uhasselt.be (M.P.M.V.); 2Department of Veterinary Management of Animal Resources, Faculty of Veterinary Medicine, Liège University, 4000 Liège, Belgium; 3Kenya Wetlands Biodiversity Research Team (KENWEB), Ichthyology Section, National Museums of Kenya, Nairobi 40658-00100, Kenya; 4Royal Belgian Institute of Natural Sciences, OD Natural Environment, Freshwater Biology, 1000 Brussels, Belgium; 5Kenya Fisheries Service, National Aquaculture Technology Development and Innovations Transfer Centre, Sagana 26-10230, Kenya; mungaindegwa@gmail.com

**Keywords:** tilapia, Upper Tana River region, metacercariae, yellow grub, trematodes

## Abstract

*Clinostomum* is a genus of parasitic trematodes found worldwide, infecting a wide range of hosts, including freshwater fishes, snails, birds and occasionally humans. In this study, clinostomid metacercariae were collected from Nile tilapia raised in fish farms in the Upper Tana River region, Kenya. The prevalence of infection was 17.2%, with metacercariae infecting the skin, gills and buccal cavity of the fish. Using light microscopy, scanning electron microscopy (SEM) and molecular methods targeting both nuclear ribosomal (ITS1, 5.8S, ITS2) and mitochondrial (COI) regions, the metacercariae were identified as *C. cutaneum*, *C. phalacrocoracis*, *C. tilapiae* and *Euclinostomum heterostomum*. The three species of *Clinostomum* have previously been reported to infect fish or piscivorous birds in Kenya, while this is the first report of *E. heterostomum* in this country. SEM analysis revealed new ultrastructural features of *C. cutaneum*, including an excretory pore surrounded by minute spiny papillae, an everted cirrus and dome-shaped papillae on the tegumental area around the genital pore. The cirrus lacked basal papillae, showing morphological variation between the adult and metacercarial stages. Our study, therefore, provides new insights into the phenotypic identification of flukes that may be pathogenic to fishes and humans and, therefore, of scientific and practical importance.

## 1. Introduction

The global production of the Nile tilapia *Oreochromis niloticus* (Linnaeus, 1758) has increased over the last thirty years due to the increasing demand for animal protein [1,2]. In Kenya, this species accounts for 80% of total aquaculture production and has gained popularity in fish farming, including in regions like Central Kenya, where fish consumption is not traditionally common [3,4,5]. Due to the increase in inland aquaculture practices, fish disease outbreaks, increased mortality, and higher parasite burdens are highly likely to occur [6], prompting attention from researchers and fishery stakeholders. In Kenya, research into fish parasitology is steadily growing; approximately 119 species of fish parasites have been reported, with only 83 identified at the species level [7].

Clinostomidae Lühe, 1901, comprises digenetic trematodes with a heteroxenous life cycle that involves multiple hosts. The adult stages are commonly found in fish-eating birds’ buccal cavities and oesophagus as definitive hosts [8,9]. Their life cycle begins when birds release eggs into aquatic environments, which subsequently hatch into free-swimming miracidia. Miracidia infect freshwater snails as the first intermediate hosts and various fish species as the second intermediate hosts, harbouring the metacercarial stages [10]. Although rarely infected, humans and mammals have occasionally been reported as accidental hosts of representatives of Clinostomidae [11,12]. Due to the increased number of reports of members of *Clinostomum* Leidy, 1856, in aquaculture conditions, it has recently been the subject of many studies. Advances in genetic research have expanded the *Clinostomum* species catalogue, addressing the challenges posed by relying solely on morphological characteristics, which often show high similarity and minimal variation between species. These genetic tools, therefore, allow for the identification of previously unrecognized species and enable linking larval stages to their corresponding adult forms [9,13,14,15,16,17].

Despite more than 50 species being suggested worldwide as members of *Clinostomum,* only 15 species are considered valid to date [9,13,18,19]. The diversity of *Clinostomum* species in the Afrotropical region, on the other hand, is insufficiently explored, with only four species currently recognized: *C. cutaneum* Paperna, 1964, *C. phalacrocoracis* Dubois, 1931, *C. tilapiae* Ukoli, 1966 and *C. ukolii* Caffara, Locke, Echi, Halajian, Luus-Powell, Benini, Tedesco & Fioravanti, 2020 [15]. Similarly, the limited understanding of the effects of these parasites on Nile tilapia and the conditions facilitating their emergence make it difficult for aquaculture stakeholders to access valuable information for informed decision-making to support aquaculture sustainability [20]. It is interesting to look at the parasitic fauna infecting fish in the central region of Kenya, as most inland aquaculture production occurs there [18]. For this reason, we conducted a survey in the Upper Tana River region with the purpose of assessing the diversity of parasites infecting Nile tilapia reared in fish farms. Regarding clinostomid infections, three species of *Clinostomum*—*C. cutaneum*, *C. phalacrocoracis* and *C. tilapiae*—have been reported previously in Nile tilapia in Kenya [13,21]. The present study identified the metacercariae using light microscopy, scanning electron microscopy (SEM) and molecular methods. For scanning electron microscopy, surface observations have revealed new ultrastructural features important for the taxonomy and systematics of a wide range of organisms [22]. The findings in this study contribute to resolving taxonomic ambiguities within *Clinostomum*. In particular, we present additional features in *C. cutaneum* that were not seen in earlier studies, further enhancing the accuracy of species differentiation within this genus.

## 2. Materials and Methods

### 2.1. Study Area and Sample Collection

The Upper Tana River region covers around 15,000 km^2^ and is characterized by the highest precipitation rates in Kenya, with a humid or semi-humid climate year-round [23]. Between mid-January and mid-February 2024, we collected 157 Nile tilapia specimens from fish farms in this area after obtaining a research permit from the National Commission for Science, Technology and Innovation (NACOSTI), permit: NACOSTI/P/23/31261. The fish hosts were then sacrificed by cervical dislocation, and a fin clip of each specimen preserved in absolute ethanol was deposited at the Royal Belgian Institute of Natural Sciences, Belgium (AB49103238-285, AB42579285-332, AB42579752-764, AB42610341-388). The map showing the sampling localities (Figure 1) was created using QGIS v3.38.3 (QGIS Development Team 2022, QGIS Information System, Open Source Geospatial Foundation Project. http://qgis.osgeo.org, accessed on 10 June 2024).

### 2.2. Parasitological Examination

The external surfaces and internal body organs of fish were carefully inspected using the naked eye and a stereomicroscope to detect encysted metacercariae. The encysted metacercariae were carefully excised using a fine needle, relaxed with boiling water, and preserved in 70% ethanol for further morphological processing. Additional specimens were preserved in absolute ethanol for subsequent molecular analysis. The infection parameters, i.e., prevalence (P) and mean intensity (M.I), were calculated according to Bush et al. [24].

### 2.3. Morphological Identification

For morphological identification, we only used a subset of the individuals isolated as some were too small to excise and use or too big to mount on slides. For this, ten specimens were stained with Borax carmine, dehydrated in a graded ethanol series for 30 min—30%, 50%, 70% (three times), 80%, 95%, and 100% (three times)—cleared with Amman’s lactophenol and mounted on permanent slides using Euparal. The metacercariae were viewed using a Leica DM2500 optical microscope (Leica Microsystems GmbH, Wetzlar, Germany) fitted with a Leica DMC4500 camera. The voucher specimens were deposited in the collection of the Research Group Zoology: Biodiversity and Toxicology at Hasselt University (HU) (Diepenbeek, Belgium) (HU XXIII.2.02-2.22).

### 2.4. Scanning Electron Microscopy

Ten specimens were prepared for scanning electron microscopy. The specimens were post-fixed in 4% osmium tetroxide (OsO_4_), thoroughly washed in distilled water to remove excess osmium and dehydrated in a graded ethanol series for 30 min each: 30%, 50%, 70% (three times), 80%, 95% and 100% (three times). They were then dried using hexamethyldisilazane under a fume hood overnight. Afterwards, they were mounted on aluminum SEM stubs with double adhesive tape and gold coated at 30 mA using a JEOL JFC-1300 (JEOL Ltd., Tokyo, Japan) sputter coater. Imaging was carried out with a Phenom XL G2 Desktop Scanning Electron Microscope (ThermoFisher Scientific Waltham, MA, USA) at an accelerating voltage of 5 kV.

### 2.5. DNA Extraction, Amplification and Sequencing

Sequences of the nuclear gene portions from internal transcribed spacer 1 (ITS1), 5.8S, ITS2, 28S rDNA and mitochondrial cytochrome *c* oxidase subunit 1 (COI mtDNA) were obtained using polymerase chain reaction (PCR). These nuclear ribosomal markers evolve at different rates, which makes them suitable for assessing genetic divergence at the interspecific level [25,26]. The COI mtDNA, on the other hand, is a promising resource for assessing intraspecific genetic differentiation because it is a fast-evolving marker compared to nuclear rDNA [27].

The posterior end of the metacercariae was cut, and DNA extraction was performed using the protocol adapted by Kmentová et al. [28]. Samples stored in 99% ethanol were spun down, ethanol was removed and they were left to dry for 30 min. Then, 195 µL of TNES buffer (400 mM NaCl, 20 mM EDTA, 50 mM Tris pH 8, 0.5% SDS) and 5 µL of Thermo Scientific^TM^ proteinase *K* (20 mg/mL) were added to the samples. After incubation at 55 °C overnight, 2 µL of Invitrogen^TM^ yeast tRNA (10 mg/mL) was added as a carrier and briefly spun down before adding 65 µL of 5 M NaCl and 290 µL of 96% ethanol. The samples were cooled for 60 min at −20 °C and then spun down for 15 min at 18,000 rcf to a small white pellet. The supernatant was removed and replaced with 1 mL of chilled 70% ethanol. The samples were centrifuged for 8 min at 18,000 rcf (this ethanol-rinsing step, removing the supernatant, adding ethanol, and centrifuging were repeated once). The supernatant was removed, and the DNA was eluted in 30 µL of 0.1 × TE buffer (0.02% Thermo Scientific^TM^ Tween-20 washing buffer). The DNA extract was placed overnight at 4 °C for resuspension and stored at −20 °C.

Partial ITS1, 5.8S, ITS2 and 28S regions were amplified using the forward primer NC5 (5′-GTA GGT GAA CCT GCG GAA GGA TCA TT-3′) [26] and the reverse primer NC2 (5′-TTA GTT TCT TTT CCT CCG CT-3′) [29]. The PCR reaction was performed using MangoMix™. For each 2 μL of DNA extract, 12.50 μL of MangoMix™, 0.50 μL of MgCl_2_ (1 mM), 1.25 μL of the forward and reverse primers (0.5 μM), respectively, and 7.50 μL of ddH_2_O was added, adding up to a total of 25 μL per reaction. The PCR amplification occurred under the following conditions: initial denaturation for 2 min at 94 °C, 39 cycles for 1 min at 94 °C, 1 min at 52 °C, and 1:30 min at 72 °C, final elongation for 7 min at 72 °C and cooling to 4 °C.

Part of the mitochondrial COI gene was amplified using forward ASmit1 (5′-TTT TTT GGG CAT CCT GAG GTT TAT-3′) and reverse ASmit2 (5′-TAA AGA AAG AAC ATA ATG AAA ATG-3′) primers, both widely used for digeneans and other flatworms [30]. For each 2 μL of DNA extract, 12.50 μL of MangoMix™, 0.50 μL of MgCl_2_ (1 mM), 1.25 μL of the forward and reverse primers (0.5 μM), respectively, and 7.50 μL of ddH_2_O were added, adding up to a total of 25 μL per reaction. The PCR conditions were set as follows: 2 min initial denaturation at 94 °C, 37 cycles of 30 s at 94 °C, 40 s at 48 °C, and 50 s at 72 °C, final elongation for 5 min at 72 °C and cooling to 4 °C. Gel electrophoresis was used to check for successful amplification. The PCR products were excised and purified using a GeneJet Purification Kit (Thermo Fisher Scientific, Waltham, MA, USA) according to the manufacturer’s guidelines. The corresponding primer pairs used for amplification were used for subsequent Sanger sequencing in Macrogen (Amsterdam, The Netherlands).

### 2.6. Sequence Analysis

We verified our morphological identification of the sequences generated in this study by BLASTing [31] the obtained sequences on the NCBI website, which allowed us to identify the closest congeners. Multiple sequence alignments were constructed using MUSCLE v5 [32] under default parameters, and maximum-likelihood-based model selection was performed in Molecular Evolutionary Genetics Analysis (MEGA) v11.0.13 [33]. After choosing a model in MEGA based on the Bayesian Information Criterion, we selected the highest-ranked model available in this software to calculate pairwise distances. As a result, we used the Kimura 2-parameter model [34] for the ITS sequences and the Tamura-Nei model [35] for the COI sequences. A haplotype genealogy graph was constructed using Fitchi [36], using all available sequences of the closest congeneric species. All sequences were submitted to GenBank (PV123689-PV123701).

## 3. Results

### 3.1. Infection Parameters and Isolated Metacercariae

Among the 157 examined hosts, 27 individuals were found to be infected with metacercariae (Figure 2D); using a subset of the isolated individuals, we identified them to belong to a total of four species: three species of *Clinostomum* (*C. cutaneum*, *C. phalacrocoracis* and *C. tilapiae*) and one species of *Euclinostomum* (*E. heterostomum*) (Appendix A). Due to practical constraints, we could not provide prevalence estimates for all the species. Some metacercariae were too small to examine or too large to be mounted on slides for detailed analysis. The prevalence of infection was 17.2% (95% CI: 11.3–23.1%), while the mean intensity was 7.3. The minimum number of metacercariae collected from each host was 1, while the highest was 38. Mixed infections involving *C. cutaneum* and *C. phalacrocoracis*, *C. phalacrocoracis* and *C. tilapiae*, and *C. cutaneum* and *E. heterostomum* were observed. Most of the clinostomid metacercariae were recovered from the skin (infection frequency = 0.54) of the infected hosts (Figure 2C), followed by the buccal cavity (infection frequency = 0.45) (Figure 2A) and occasionally the gills (infection frequency = 0.01) (Figure 2B), as shown in Figure 2.

### 3.2. Morphological Observations in *C. cutaneum*

The morphological features of eight of our specimens are consistent with the characterization of *C. cutaneum* previously reported by Gustinelli et al. [9]. The key observations include a distinct Y-shaped uterus, intestinal caeca extending laterally from the anterior to the posterior body ends and a smaller anterior testis than the posterior testis (Figure 3).

### 3.3. Scanning Electron Microscopy Results

Scanning electron microscopy of *C. cutaneum* in this study revealed several novel features. The excretory pore (Figure 4E,F), previously undetected in the work of Gustinelli et al. [9] due to a possible cuticular fold, was clearly visible and surrounded by minute, spiny papillae (white arrow in Figure 4E). Additionally, an everted cirrus (Figure 4C) was observed. In the adult stages of *C. cutaneum*, Gustinelli et al. [9] observed basal papillae in the cirrus. However, the cirrus observed in our study lacked basal papillae (encircled in red), highlighting morphological differences between the adult and metacercarial stages. The tegumental area around the genital pore (Figure 4D) was also surrounded by dome-shaped papillae (white asterisks) never reported before, further enriching the morphological characterization of this species.

### 3.4. Molecular Analyses

Thirteen partial ITS1-5.8S-ITS2 sequences were obtained in this study, ranging in length from 1043 to 1102 bp. These sequences included 554–570 bp corresponding to ITS1, 157 bp corresponding to 5.8S and 290–328 bp corresponding to ITS2. Eight newly generated sequences (GenBank accession numbers PV123689-91, PV123693-95, PV123697, PV123699) showed an average similarity of 99.80% (range: 99.88–100%) to four published sequences of *C. cutaneum* (Table 1) published in [9,14]. Three sequences (GenBank accession numbers PV123696, PV123698, PV123700) averaged 99.9% similarity (range: 99.94–100%) to six sequences of *C. phalacrocoracis* published in [9,14,37] (KP110567, KP110567, FJ609422-23, KJ786975-76). One sequence (GenBank accession number PV123701) averaged 99.7% similarity (range: 99.59–99.7%) to eight sequences of *C. tilapiae* published in [15] (KY649349-55), while one sequence (GenBank accession number PV123692) averaged 99.9% similarity (range: 99.16–100%) to eleven sequences of *E. heterostomum* published in [38] (KP721422-25, KP721427, KP721430-31, KP721435, KP721437-39).

A total of 43 sequences of congeners, all identified to the species level, were obtained from GenBank (Table 1) for further sequence analyses.

We provide the intraspecific and interspecific differences in ITS sequences in Table 2.

The percentage identity of the eight COI mtDNA sequences (579 bp) obtained in this study was not comparable to other sequences in GenBank using the online BLAST tool available at the NCBI website (http://blast.ncbi.nlm.nih.gov/Blast.cgi, accessed on 1 February 2025). This is because the primers used in this study [30] amplified a region different from the published sequences for clinostomids (positions 1–688 bp in the COI mtDNA gene). The sequences obtained in this study overlapped the region between 699 and 1287 bp in the COI mtDNA gene from the mitogenome of *C. complanatum* (OP681143) [39]. The pairwise distances of COI mtDNA generated in this study are shown in Table 3.

Based on the values provided in Table 3, the isolates from the same species showed very low genetic distances (≤0.009), indicating high intraspecific similarity within the species. The intraspecific variation in *C. cutaneum* for COI (0.000–0.009) is slightly larger than ITS rDNA (0.000–0.002), suggesting that COI might show greater variability within this species. For *C. phalacrocoracis,* both markers show minimal intraspecific variation. The COI gene shows complete uniformity (0 differences, pairwise distance: 0.000), whereas its ITS rDNA shows minor variation.

For the haplotype networks of the 807 bp region covering the ITS1, 5.8S and ITS2 regions, we used the minimum spanning tree model [40] to visualize the genetic structure in the different populations and species (Figure 5). The haplotype of *C. cutaneum* (black, present study) is shared with those previously described in Kenya (yellow). Similarly, *C. phalacrocoracis* (blue) shared the same haplotype with Kenyan (black, present study) and Israeli (light blue) populations, suggesting gene flow or dispersal between these regions. The *C. tilapiae* haplotype (black, present study) is shared with the Nigerian haplotype (pink). For *E. heterostomum* (green), one haplotype is shared with published sequences from Israeli populations. However, three additional haplotypes are observed, suggesting intraspecific genetic diversity within this species despite all isolates being collected from the same region in Israel.

## 4. Discussion

To date, 15 species of *Clinostomum* have been identified using a combination of molecular and morphological methods [18], which provide a reliable basis for diagnosis within this genus [9,13,15,41]. As part of a comprehensive study aimed at surveying the diversity of parasites infecting Nile tilapia in the Upper Tana River region in Kenya, we identified the isolated metacercariae as *C. cutaneum, C. phalacrocoracis, C. tilapiae* and *E. heterostomum*. The first three species have previously been reported to infect Nile tilapia or grey herons in Kenya. The report of *C. tilapiae* in this region lacked supporting morphological or molecular data. This study provides so far lacking molecular data for *C. tilapiae* and *E. heterostomum* in Kenya, but since only one specimen was available for each species, no morphological characterizations of these metacercariae were possible. For this reason, more work still needs to be carried out to properly characterize the morphology of *E. heterostomum* and *C. tilapiae* in *O. niloticus* (a cichlid host), with the latter parasite species having a previous report with genetic data from the mochokid catfish *Synodontis batensoda* Rüppell, 1832 in Nigeria.

Regarding infection parameters, a high prevalence of *Clinostomum* spp. was observed in cichlids in Lake Kinneret, Israel (23.4%) [37], in cultured *O. niloticus* in Sahary fish hatchery, Egypt (25%), in wild *Sarotherodon galilaeus* (Linnaeus, 1758) in Lake Nasser, Egypt (33%) [42], and our study (17.2%). Similarly, research by Mahdy et al. [43] found that farmed fish had a higher infection rate (32%) than wild tilapia (24%), suggesting that *Clinostomum* infections may be widespread in aquaculture systems. Higher parasite burdens in farmed fish can lead to reduced market value and increased mortality, particularly in cases where heavy infestations cause damage to fish tissues, gills and skin [16,42,44]. The ability of metacercariae to impair host health through secondary infections from bacteria or fungi, ultimately resulting in death, has been documented [42,45,46], reinforcing the need for proactive parasite management strategies in tilapia farming.

While SEM helps visualize detailed structural characteristics of the tegument and surface morphology that are otherwise missed with light microscopy, it has only been applied to some species within *Clinostomum*. Compared to the surface ultrastructure of *C. ukolii* and *C. tilapiae*, whose teguments are completely covered with minute spines, the tegument of *C. cutaneum* lacks spines. Similarly, dome-like structures were observed only on the genital pore of *C. cutaneum*, whereas in *C. ukolii,* such structures are present across the tegumental surface and sometimes between suckers [17]. In the present study, we observed additional diagnostic features previously not mentioned by Gustinelli et al. [9].

## 5. Conclusion

This study provides new diagnostic features for the metacercariae of *C. cutaneum,* refining its morphological characterization. It also provides the first molecular data for *C. tilapiae* in Kenya, reports the first species occurrence for *E. heterostomum* in Kenya, and confirms its presence on Nile tilapia in this country. Additionally, we offer a molecular barcoding resource in the form of COI mtDNA sequences, with our selected primers amplifying a previously unsequenced region of the mitochondrial genome. The high prevalence of *Clinostomum* spp. shows their potential impact on both fish and human health, emphasizing the need for continued monitoring. Finally, we demonstrate the value of SEM as a complementary tool for more precise parasite species identification.

## Figures and Tables

**Figure 1 pathogens-14-00249-f001:**
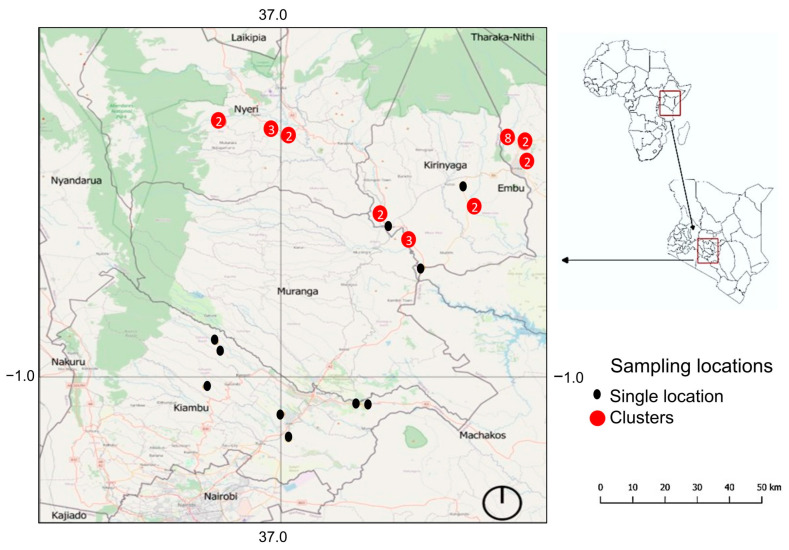
Map showing sampling localities of Nile tilapia examined for the presence of clinostomid metacercariae.

**Figure 2 pathogens-14-00249-f002:**
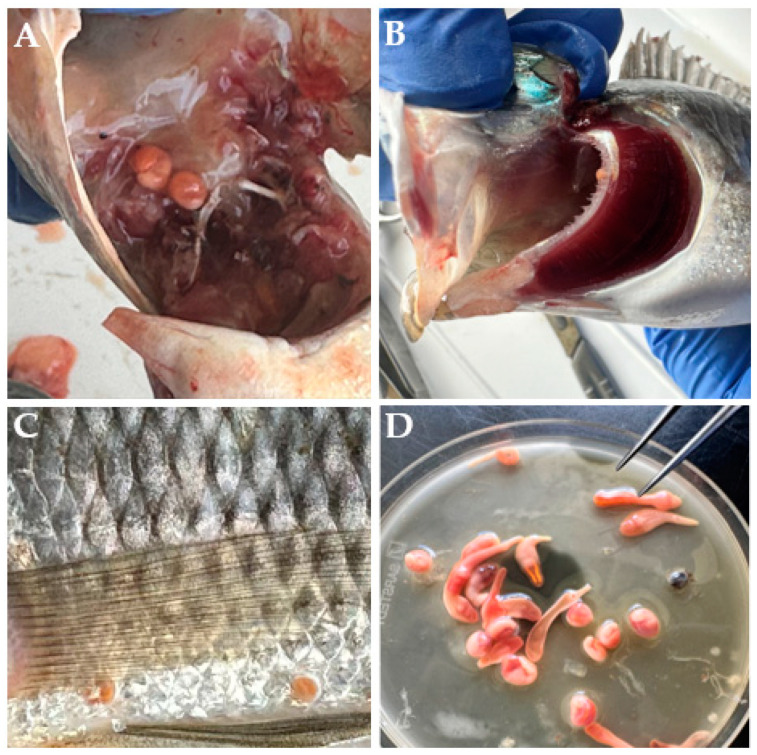
Metacercarial infection on the buccal cavity (**A**), gills (**B**) and skin (**C**) of Nile tilapia; (**D**) isolated metacercariae.

**Figure 3 pathogens-14-00249-f003:**
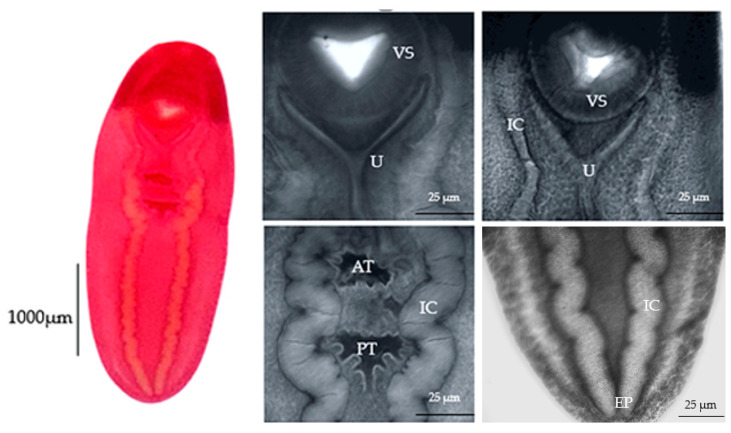
Light photomicrographs of *Clinostomum cutaneum* in the present study. VS: ventral sucker; U: uterus; AT: anterior testis; PT: posterior testis; EP: excretory pore; IC: intestinal caeca.

**Figure 4 pathogens-14-00249-f004:**
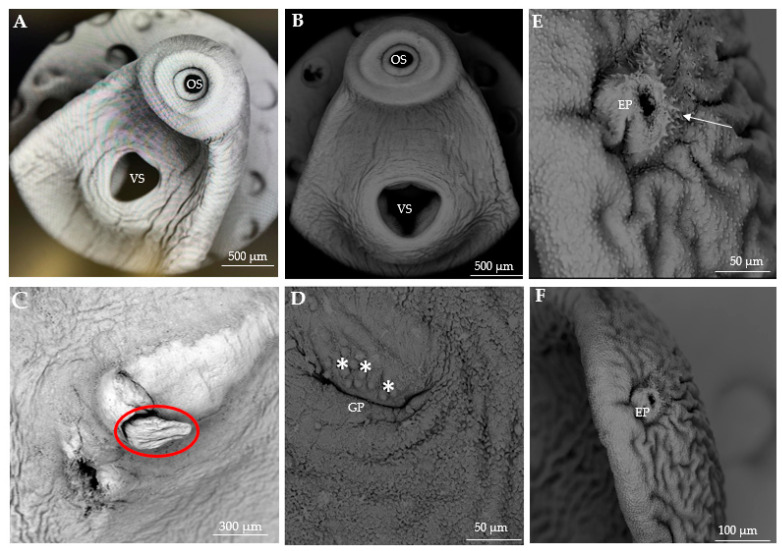
Scanning electron micrographs of *Clinostomum cutaneum*. (**A**,**B**) Anterior end showing the oral sucker (OS) and ventral sucker (VS). (**C**) Everted cirrus (circled in red). (**D**) Genital pore (GP) surrounded by dome-shaped papillae (white asterisks). (**E**,**F**) Excretory pore (EP) on the posterior end (white arrow in (**E**)).

**Figure 5 pathogens-14-00249-f005:**
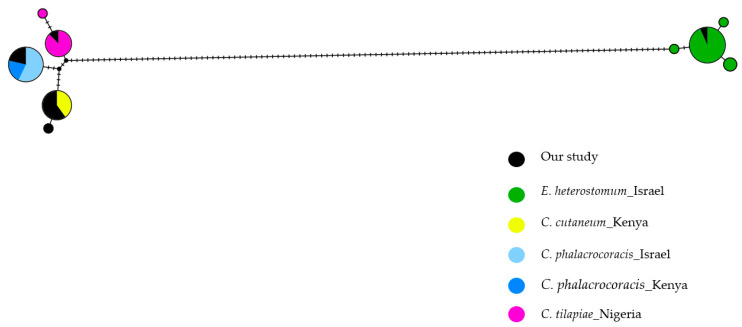
Haplotype genealogy graph based on ITS rDNA (807 bp) fragments of *C. cutaneum*, *C. phalacrocoracis*, *C. tilapiae* and *E. heterostomum*. The colours represent different species and geographic locations. The total Fitch distance is 129 substitutions, reflecting the minimum number of inferred mutations under the Fitch parsimony criterion. The node sizes are proportional to the haplotype frequencies, and small crossbars along the branches denote mutational steps between haplotypes.

**Table 1 pathogens-14-00249-t001:** List of published sequences included in genetic analysis.

Parasite Species	Host Species	Locality	Accession Number	Reference
*C. cutaneum*	*Oreochromis niloticus*	Sagana, Kenya	PV123689-91, PV123693-95, PV123697, PV123699	Present study
*C. phalacrocoracis*	*Oreochromis niloticus*	Sagana, Kenya	PV123696, PV123698, PV123700	Present study
*C. tilapiae*	*Oreochromis niloticus*	Sagana, Kenya	PV123701	Present study
*E. heterostomum*	*Oreochromis niloticus*	Sagana. Kenya	PV123692	Present study
*C. cutaneum*	*Oreochromis niloticus*	Sagana, Kenya	KP110564-65	[14]
*C. cutaneum*	*Oreochromis niloticus*	Sagana, Kenya	FJ609421	[9]
*C. cutaneum*	*Ardea cinerea*	Sagana, Kenya	GQ339114	[9]
*C. phalacrocoracis*	*Ardea cinerea*	Sagana, Kenya	FJ609423	[9]
*C. phalacrocoracis*	*Oreochromis niloticus*	Sagana, Kenya	FJ609422	[9]
*C. phalacrocoracis*	*Oreochromis niloticus*	Sagana, Kenya	KP110567-69	[14]
*C. phalacrocoracis*	Cichlids	Lake Kinneret, Israel	KJ786975-982	[37]
*C. tilapiae*	*Synodontis batensoda*	Anambra basin, Nigeria	KY649349-356	[15]
*E. heterostomum*	Cichlids	Lake Kinneret, Israel	KP721422-439	[38]

**Table 2 pathogens-14-00249-t002:** Number of nucleotide differences and model-corrected pairwise distances between the newly generated sequences of ITS rDNA and previously published sequences. The values above the diagonal (in grey) for interspecific differences indicate the range of nucleotide differences, while those below the diagonal indicate the range of distances.

Interspecific					Intraspecific	
	*C. tilapiae*	*C. cutaneum*	*C. phalacrocoracis*	*E. heterostomum*	# Differences	Distance
*C. tilapiae*		7–11	6–10	120–126	0–3	0.000–0.006
*C. cutaneum*	0.008–0.015		7–9	121–125	0–1	0.000–0.002
*C. phalacrocoracis*	0.006–0.012	0.004–0.014		122–126	0–1 *	0.000
*E. heterostomum*	0.169–0.215	0.203–0.220	0.199–0.220		0–3	0.000–0.014

* This difference is masked completely in the haplotype network analysis due to unambiguous bases.

**Table 3 pathogens-14-00249-t003:** Model-corrected pairwise distances of eight COI mtDNA sequences generated in this study.

Interspecific			Intraspecific	
	*C. cutaneum*	*C. tilapiae*	# Differences	Distance
*C. cutaneum*			0–1	0.000–0.009
*C. tilapiae*	0.114–0.131		-	-
*C. phalacrocoracis*	0.124–0.143	0.081–0.127	0	0.000

## Data Availability

Voucher materials from this study have been deposited in the Zoology: Biodiversity and Toxicology collection at Hasselt University (Diepenbeek, Belgium) (catalogue number HU XXIII2.02-XXIII2.22), and host fin clips have been deposited in the Royal Belgian Institute of Natural Sciences (AB49103238-285, AB42579285-332, AB42579752-764, AB42610341-388). Additionally, the genetic sequences generated in this study have been made publicly available on GenBank under accession numbers PV123689-PV123701.

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
