# Peer review of "Scanning Electron Microscopy Reveals New Ultrastructural Features in Metacercariae of Clinostomum cutaneum (Digenea: Clinostomidae) Infecting Oreochromis niloticus (Actinopterygii: Cichlidae) in Kenya"

_pathogens, 2025, doi:10.3390/pathogens14030249_

Round 1

Reviewer 1 Report

Comments and Suggestions for Authors

Dear authors, here are some comments and suggestions:
- Line 102: Why was the deposit code not included?
- Line 106: Replace ":" with "." after the term Figure. Make Figure bold. The same should be done in the captions of the other figures and tables.
- In item 3.1, also include the confidence interval of the prevalence and the prevalence of each species found.
- Line 223: Do not abbreviate the scientific name in the caption.
- In item 3.4, several deposit codes were also not mentioned.
- Why is there morphological data for only one of the reported species? Why is there no comparison between the morphological characteristics of the species? It would be very interesting to be able to compare the species morphologically, in addition to the molecular differences.
I hope that the suggestions will produce the best version of this study, which has representative and very valuable data.

Author Response

Dear reviewer,

Please find the attached PDF file addressing the comments and suggestions you provided. 

Kind regards,

Miriam.

Reviewer 2 Report

Comments and Suggestions for Authors

In section 3.1 Infection Parameters and Metacercariae Isolate, the authors should provide a table with data on prevalence and infection frequency. Additionally, the 95% confidence interval (CI95%) should be calculated.

In sections 3.2 Morphological Observations in C. cutaneum and 3.3 Scanning Electron Microscopy Results, the authors should avoid commenting on the results. Please be concise and focus solely on the findings. The same pattern of presenting results was also used in section 3.4 Molecular Analyses.

In the Results section, it is not customary to compare results with those of other studies. Therefore, my suggestion for this section is to focus solely on describing the obtained results. Alternatively, if the authors agree, I would like to recommend merging the Results and Discussion sections into a single section.

Author Response

(The authors gave the same response as above.)
